# Impact of the COVID-19 Pandemic on the Patient’s Decision about Bariatric Surgery: Results of a National Survey

**DOI:** 10.3390/medicina57080756

**Published:** 2021-07-26

**Authors:** Maciej Walędziak, Anna Różańska-Walędziak, Paweł Bartnik, Joanna Kacperczyk-Bartnik, Andrzej Kwiatkowski, Michał Janik, Piotr Kowalewski, Piotr Major

**Affiliations:** 1Department of General, Oncological, Metabolic and Thoracic Surgery, Military Institute of Medicine, 04-141 Warsaw, Poland; maciej.waledziak@gmail.com (M.W.); akwiatkowski@wim.mil.pl (A.K.); mjanik@wim.mil.pl (M.J.); pkowalewski@wim.mil.pl (P.K.); 22nd Department of Obstetrics and Gynecology, Medical University of Warsaw, 00-315 Warsaw, Poland; bartnik.pawel@gmail.com (P.B.); asiakacperczyk@gmail.com (J.K.-B.); 32nd Department of General Surgery, Jagiellonian University Medical College, 31-008 Kraków, Poland; piotr.major@uj.edu.pl

**Keywords:** bariatric surgery, COVID-19, COVID-19 lockdown

## Abstract

*Background:* the COVID-19 pandemic and the implemented restrictions have changed the functioning of healthcare systems worldwide. The purpose of the study was to evaluate the impact of the present epidemiological situation on patients’ decisions about undergoing weight loss surgery. *Methods:* data were collected from 906 bariatric patients by the means of a national online survey, the majority of whom were women (87.9%). The survey started on 9 April 2020 and was open until 28 April 2020. The questionnaire included multiple choice and open questions, divided into three chapters: general information about the patient, life during the COVID-19 pandemic, and bariatric care during the COVID-19 pandemic. *Results:* despite the pandemic and the associated risk of COVID-19 infection, 443 responders (48.9%) would have decided to undergo bariatric surgery. Awareness of the negative impact of obesity on the course of COVID-19 illness had only marginable impact on patients’ decision-making (76.6% vs. 75.3%; *p* < 0.80). Contact with COVID-19 prior to the survey had a negative impact on the willingness to undergo bariatric surgery (3.0% vs. 4.4%; *p* < 0.55). There was a positive correlation between the BMI and preference for bariatric surgery in the time of the pandemic (37.4 ± 9.0 vs. 34.9 ± 8.7; *p* < 0.001). *Conclusions:* the level of awareness about the advantages of operative treatment of obesity is high among bariatric patients. The majority of patients awaiting bariatric surgery at the moment of the survey were positive about undergoing bariatric surgery despite the increased risk of a serious course of COVID-19 infection. Therefore, a large proportion of patients was determined to have bariatric treatment even during the pandemic, being aware of the increased risk of worse pace of COVID-19 disease in case of obesity and related diseases.

## 1. Introduction

Since the beginning of the COVID-19 pandemic, it has been the major global health concern. More than 100 million of cases have been detected worldwide and more than 2,000,000 people have died due to COVID-19 infection [1].

The pandemic forced the reorganization of healthcare systems all over the world. Since the World Health Organization (WHO) declared the COVID-19 pandemic on 11 March 2020, most countries around the world have implemented epidemiological restrictions and redirected their resources to fight the pandemic [2,3]. One of the consequences of the situation was the suspension of elective surgery, except for oncological procedures and life-saving operations [4]. Bariatric operations were withdrawn in most countries, following the recommendations of the International Federation for the Surgery of Obesity and Metabolic Disorders (IFSO) [5]. About 60,000 bariatric operations per month might have been canceled worldwide [6]. After a few months, some countries have started lowering the level of restrictions, including reopening shops, public places, allowing social intercourse, and returning to performing elective surgery.

The aim of this study was to investigate the impact of the COVID-19 pandemic on patient’s decisions about undergoing bariatric surgery.

## 2. Materials and Methods

This study includes data collected by the means of a national internet survey conducted among pre- and post-bariatric surgery patients via Google Forms. The survey started on 9 April 2020 and was open until 28 April 2020. The online survey was published and distributed via social media (Facebook) among members of the Polish Bariatric Patients’ Association (CHLO). The number of CHLO members is around 8100; however, we could not evaluate the real number of active members, who had read information about the survey. Therefore, we could not establish the actual response rate. The questionnaire included multiple choice and open questions, divided into three chapters: general information about the patient, life during the COVID-19 pandemic, and bariatric care during the COVID-19 pandemic. The survey is attached as Table A1. The project was supported by the Metabolic and Bariatric Chapter of Polish Surgeons’ Association (SCMiB). The data was completely anonymized and contained no patient identification data. The questionnaire was evaluated and approved by several independent experts in the field of bariatric surgery.

### 2.1. Statistical Analysis

Results are presented as means with standard deviations or medians with interquartile ranges.

We performed the statistical analysis using Statistica 13 (StatSoft Inc., Tulsa, OK, USA). Normality of the data was tested with the Shapiro–Wilk test. Continuous variables were compared with the Student’s t test for normally distributed or the Mann–Whitney U test for non-normally distributed data. Categorical variables were compared using the chi2 or Fisher test. Statistical significance was set at *p* < 0.05.

### 2.2. Ethical Considerations

The study was anonymous and performed in accordance with the ethical standards laid down in the 1964 Declaration of Helsinki and its latter amendments (Fortaleza). Participants were informed about the aim of the study and informed consent was obtained electronically prior to the beginning of the survey. The study was approved by the Bioethics Committee of the Jagiellonian University (1072.6120.103.2020). 

## 3. Results

In this study, 906 patients were included, and most of them were female (87.9%). Due to this discrepancy of proportion between the sexes, the results of our study are more applicable to female patients. Five hundred ninety-six participants were after bariatric surgery (65.8%). The median BMI was 36.1 (±8.9). Most patients reported at least one comorbidity, the most common of which was osteoarthritis, followed by hypertension, insulin resistance, type 2 diabetes mellitus, obstructive sleep apnea, and dyslipidemia.

The demographic characteristics of the study population are presented in Table 1.

Despite the COVID-19 pandemic and the risk of infection, 443 responders (48.9%) would decide to undergo bariatric surgery in the current situation if they were given the choice. People who had personal contact with relatives or friends potentially infected with COVID-19 (6.1% vs. 6.9%, *p* < 0.61) and patients aware of the negative impact of obesity on the course of COVID-19 illness were less positive about undergoing surgery during the pandemic (70.8% vs. 75.5%, *p* < 0.12). A significantly higher proportion of men than women would decide positively about bariatric treatment at present (15.23% vs. 9.11%; *p* < 0.006). There was a positive correlation between the body mass index (BMI) and the willingness to undergo bariatric surgery (37.4 ± 9.1 vs. 34.9 ± 8.7; *p* < 0.001).

Table 2 and Table 3 present patients’ opinion about undergoing bariatric surgery during the COVID-19 pandemic depending on sex, age, BMI, and comorbidities.

Among the group of patients before bariatric surgery, 206 (66.5%) patients were willing to undergo the surgery despite the pandemic. Respondents who had had contact with people potentially infected with COVID-19 were more negative about surgery during the pandemic (3.0% vs. 4.4%; *p* < 0.55). Awareness of the negative impact of obesity on the severity of the course of COVID-19 illness had only a marginable impact on decision making (76.6% vs. 75.3%; *p* < 0.80). Hypertension had a statistically significant positive influence on the decision about bariatric treatment during the pandemic (27.7% vs. 39.2%; *p* < 0.05.)

Table 3 presents pre-operative patients’ opinion about undergoing bariatric surgery during the COVID-19 pandemic depending on sex, age, BMI, and comorbidities.

Significantly more pre-operative patients preferred not to undergo surgery if they were aware that their bariatric center was in a hospital responsible for treatment of COVID-19 patients (38.4% vs. 63.4%; *p* < 0.003). The group of respondents who felt more anxiety or fear about their health or life due to the pandemic also preferred to delay their bariatric treatment (72.1% vs. 84.5%; *p* < 0.03).

## 4. Discussion

Our survey showed that there are many factors influencing the patients’ decision about undergoing bariatric surgery during the pandemic.

Despite international recommendations to postpone elective surgery until after the epidemic, [5] there is a large proportion of patients who would like to undergo bariatric surgical treatment despite the risk of developing COVID-19 illness. Nearly half of all bariatric patients were positive about having bariatric surgery during the COVID-19 pandemic. The proportion was even higher in the group of pre-operative patients, with more than two thirds of pre-operative patients having wanted to undergo surgery despite the higher risk of a severe coronavirus infection course.

Women, who comprised the majority of our study group, are less susceptible to viral infections based on a better innate immunity, levels of steroid hormones, and factors related to sex chromosomes. [7] Despite the similar incidence of COVID-19 infection in both sexes, men have a worse prognosis. [8] Because of the unbalanced proportion of sexes in our study, it was unfortunately not possible to draw statistically significant conclusions. 

A higher number of comorbidities, especially type 2 diabetes mellitus and osteoarthritis had a positive correlation with a decision about bariatric surgery in the state of the pandemic. Contrarily, when considering the group of pre-operative patients, patients with a preference for surgery during the pandemic had fewer comorbidities and less often suffered from type 2 diabetes, insulin resistance, or osteoarthritis. Both patients scheduled for bariatric surgery in a COVID-19 treating center and those who felt more anxiety or fear due to the pandemic were negative about operative treatment during the pandemic. Although lockdown and isolation undoubtedly negatively affected dietary regimens and daily physical exercise and often resulted in an increase in body weight, these factors did not significantly influence the decision of undergoing bariatric surgery in time of increased epidemiological risk. Only a small number of patients reported health deterioration during the pandemic, but this fact did not affect their decision about the surgery.

Obesity is known to increase the vulnerability to infections, and there are already studies suggesting that a higher BMI is associated with a higher risk of COVID-19 infection, impaired treatment, and higher mortality. [9,10,11] In our study, patients with a higher BMI were more likely to decide positively about the operation despite the pandemic; contrarily, the subgroup of pre-operative patients had an opposite opinion.

Simonnet et al. reported a higher prevalence of obese patients in the group with severe acute respiratory syndrome due to COVID-19 who required invasive mechanical ventilation. [12] In our study, about 75% of patients who were positive about undergoing operation during pandemic were aware of worse the postoperative course in the case of coronavirus infection.

Telemedicine allows to reduce personal contact and decreases the risk of COVID-19 infection transmission, and it bloomed during COVID-19 pandemic. [13] In our study, more than half of patients before bariatric surgery had remote access to the doctors providing bariatric treatment and profited from online support groups during the pandemic, although these possibilities did not influence their decision about bariatric surgery during the current epidemiological situation.

It has taken China around 100 days to start getting back to “normal,” and early signals from other countries have followed, seemingly indicating a slowdown of the pace of the pandemic. [14] However, in most countries, the morbidity is still rising and it cannot be predicted how long the pandemic and the consequent restrictions on the elective surgery will last.

### Limitations of the Study

The possible limitation of our study can be the recall bias and the subjectivity of patients’ opinions. Another limitation was that the survey was conducted only among Polish bariatric patients who were able to fill it by means of internet; therefore, there was a higher proportion of participants who had a better education background and economic situation that allowed them to participate in an online survey. The high proportion of women who filled in the questionnaire (87.9%) also influenced the results, and the conclusions of our study may be correct for the specific groups of patients, female or non-poverty patients. Distribution via social media excluded the possibility of controlling the respondents or calculating the overall response rate. However, there was no incentive to introduce dishonesty into responses.

## 5. Conclusions

Our study showed that patients’ opinion about undergoing bariatric surgery during the COVID-19 pandemic is not homogenous and depends on different factors. Due to unbalanced proportion between female and male patients in our study group, our findings are mostly applicable to women, and those who have access to Internet—therefore, those who have better economic conditions. Bariatric surgery presents the only chance of recovery for a large group of patients, to help them to achieve remission of comorbidities and to reduce body weight. The level of awareness about the advantages of operative treatment is high among bariatric patients. Therefore, a large proportion of patients, in our study mostly female patients,) were determined to have bariatric treatment even during the pandemic, even though they were aware of the increased risk of a severe course of COVID-19 illness due to obesity and related diseases.

## Figures and Tables

**Table 1 medicina-57-00756-t001:** Baseline characteristics of the study population.

Variable	Total
*N*	906
Median age, years (SD)	39.1 (±8.7)
Male/Female, *n* (%)	109/789 (12.1%/87.9%)
Median BMI, kg/m^2^ (SD)	36.1 (±8.9)
Patients before/after bariatric surgery, *n* (%)	310/596 (34.2%/65.8%)
Osteoarthritis and joint pain, *n* (%)	300 (33.1%)
Hypertension, *n* (%)	294 (32.5%)
Insulin resistance, *n* (%)	291 (32.1%)
Type 2 diabetes mellitus, *n* (%)	102 (11.3%)
Obstructive sleep apnea, *n* (%)	77 (8.5%)
Dyslipidemia, *n* (%)	73 (8.1%)
Co-morbidities, *n* (IQR)	1 (0–2)

*N*: number of patients; SD: standard deviation; BMI: body mass index; IQR: interquartile range

**Table 2 medicina-57-00756-t002:** Decision about bariatric surgery during the COVID-19 pandemic—all patients.

	Would You Decide to Undergo Bariatric Surgery during the COVID-19 Pandemic?		
	Yes	No	*p*-Value
Male, *n* (%)	67 (15.2%)	38 (9.1%)	
Median age, years (SD)	39.0 ± 8.0	38.7 ± 9.0	*p* < 0.63
Median BMI, kg/m^2^ (SD)	37.4 ± 9.1	34.9 ± 8.7	*p* < 0.001
Type 2 diabetes mellitus, *n* (%)	57 (12.9%)	38 (9.1%)	*p* < 0.08
Insulin resistance, *n* (%)	138 (31.2%)	141 (33.6%)	*p* < 0.45
Arterial hypertension, *n* (%)	134 (30.3%)	145 (34.5%)	*p* < 0.18
Obstructive sleep apnea, *n* (%)	42 (9.5%)	31 (7.4%)	*p* < 0.27
Osteoarthritis, *n* (%)	159 (35.9%)	125 (29.8%)	*p* < 0.06
Dyslipidemia, *n* (%)	38 (8.6%)	33 (7.9%)	*p* < 0.71
Co-morbidities, *n* (IQR)	1.28 ± 1.2	1.22 ± 1.2	*p* < 0.51

SD: standard deviation; BMI: body mass index; IQR: interquartile range

**Table 3 medicina-57-00756-t003:** Decision about bariatric surgery during the COVID-19 pandemic—pre-operative patients.

	Would You Decide to Undergo Bariatric Surgery During the COVID-19 Pandemic?		
	Yes	No	*p*-Value
Male, *n* (%)	25 (12.2%)	14 (14.6%)	*p* < 0.57
Median age, years (SD)	38.1 ± 8.2	37.4 ± 8.8	*p* < 0.48
Median BMI, kg/m^2^ (SD)	43.5 ± 7.7	44.6 ± 7.3	*p* < 0.026
Type 2 diabetes mellitus, *n* (%)	26 (12.6%)	7 (7.2%)	*p* < 0.16
Insulin resistance, *n* (%)	76 (36.9%)	36 (37.1%)	*p* < 0.97
Arterial hypertension, *n* (%)	57 (27.7%)	38 (39.2%)	*p* < 0.05
Obstructive sleep apnea, *n* (%)	19 (9.2%)	11 (11.3%)	*p* < 0.57
Arthritis/Joint pain, *n* (%)	84 (40.8%)	35 (36.1%)	*p* < 0.44
Dyslipidemia, *n* (%)	18 (8.7%)	9 (9.3%)	*p* < 0.88
Co-morbidities, *n* (IQR)	1.4 ± 1.1	1.4 ± 1.2	*p* < 0.88

SD: standard deviation; BMI: body mass index; IQR: interquartile range

## Data Availability

The data presented in this study are available on request from the corresponding author. The data are not publicly available.

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
