# Peer review of "Impact of the COVID-19 Pandemic on the Patient’s Decision about Bariatric Surgery: Results of a National Survey"

_medicina, 2021, doi:10.3390/medicina57080756_

Round 1

Reviewer 1 Report

Summary

The manuscript with the title ‘Impact of the COVID-19 pandemic on the patient’s decision about bariatric surgery: results of a national survey’ reported a study performing an online survey to evaluate the impact of the present epidemiological situation on their decisions about undergoing weight loss surgery. The design of the study is good but the data/analysis/discussion are somehow bias. My biggest concerns are from two facts, 1) most of the patients participating the survey are female, 2) the online survey may lead to the bias data and results contributed by the biased group of communities with more knowledge about the online survey and the access to internet. The conclusions made by authors might be correct for the specific groups of patients, e.g. female patients or non-poverty patients.

Comments:

  1. Table 1, row 2 indicated that there 906 patients participating in the survey. The row 4 shows the numbers of males and females are 109 and 789, respectively. Why the numbers (109 + 789 = 898) of males plus females is not 906?
  2. Table 1, the sum of rows 7,8,9,10,11,12,13 is much greater than 906, does it mean many of 906 patients have more than one of 7 conditions (Osteoarthritis and joint pain, Hypertension, Insulin resistance, Type 2 diabetes mellitus, Obstructive sleep apnea, Dyslipidemia, Co-morbidities )?
  3. Table 2, row 3, why the sum (440 + 417 = 857) of Yes (440) and No (417) is not 906? Why the total number (105) of Males answering Yes (67) and No (38) is not 109 as showed by the row 4 in Table 1?
  4. Table 2, rows 6 - 11, why the sum (863) of Yes (443) and No (420) is not 906?
  5. Table 3, row 3, please explain what the numerator and denominator (25/205 and 14/96) The authors can put the details about each numerator and denominatorin the legend of Table 3.  Same concern applies to rows 6 - 11.
  6. Table 4, please explain what thenumerator and denominator (e.g. 10/307, 6/199, 4/91) are in each row.
  7. In section 4 Discussion, line2 129-131, the authors claimed ‘In our study, among all the responders men were more likely to decide about undergoing bariatric surgery, but in group of pre-operative patients the result was oppo-site.’. Such conclusion is not supported by the results in the study, as the numbers of male and female participants in the study are unbalancing (male vs female is14% vs 87.86%), most of the patients participating the survey are female.
  8. Based on the unbalanced male and female participants in the survey, the findings claimed by authors in the manuscript may apply to female patients better.
  9. The study foundation is based on the online survey which implicitly introduced bias caused by biased participants who have better economic condition, better access to internet. Those biases prompt us the findings in the study might only apply to the specific groups of patients, e.g. non-poverty patients who have better economic situation, better education background and internet access.

Author Response

Dear Reviewer 1,

Thank you for reviewing our manuscript.

According to your suggestion, we emphasized the fact that most of the participants in study were female and probably non-poverty patients, including the concern in the results and conclusions sections of the manuscript. We agree that, unfortunately, an online survey is a primarily biased form of gathering data, however during the state of pandemic it was the only form allowing us to get to a higher number of patients.

We corrected the mistakes and inaccuracies we had made in the tables.

Unfortunately, not all the patients gave answers to the questions in the survey, which is the reason for numbers smaller than 906 (e.g. table 1, row 4). Additionally we had excluded from table 2 patients who had responded “I don’t know”, which is the other reason for lower sums (table 2, row 3 and 6-11).

The sum of rows in the part of table 1 about comorbidities is higher than 906 because some of the patients had more than one comorbidity.

We removed the unnecessary denominators from tables 2 and 3, as they clouded the clearness and comprehensibility of the contents.

We decided about removing table 4, as the most important data are presented in the results section of the manuscript.

Following your remark, we changed the incorrect conclusion from the discussion.

We added a paragraph about the bias of the study.

We emphasized that our conclusions may apply better to female patients, who have access to internet, have better education background and economic situation that may allow them to participate in an online survey.

Reviewer 2 Report

This study aims to characterize the impact of the early phase of the COVID pandemic on the patient willingness to undergo elective bariatric surgery in Poland.

This study has several major concerns.

First the English needs significant improvement and there are formatting mistakes throughout.

Statistical output in terms of number of decimal places in inconsistent.  Numbers would only be significant to 1 decimal place.

There is no evidence that this study design was reviewed by an ethics research board prior to commencement. The appendix does not contain the informed consent document.

There is no indication of the overall response rate. There is no indication of complete response rate.

Google forms for a survey platform has no ability to prevent multiple submissions from the same source.

Author Response

Dear Reviewer 2,

Thank you for reviewing our manuscript.

We corrected the level of English language used in the manuscript and excluded the formatting mistakes.

According to your remark, we corrected the numbers used in the manuscript to 1 decimal place only.

We added information about the Bioethical Committee consent.

The overall response rate was 906 respondents out of 8100 members of the Polish Bariatric Patients’ Association. However, we cannot evaluate the actual response rate, as not all of them may be active, we can only present the overall number of members. We including this information in the study.

We agree that Google Forms surveys allow multiple submissions; however, we had no basis to suspect dishonesty among the respondents.

Round 2

Reviewer 1 Report

The authors addressed the main concerns from the previous review, the revised version of the manuscript appears to be good. It looks ready for publication as far as I can tell.

Reviewer 2 Report

Thank you for addressing the concerns raised. I have no further concerns.